# Palmitoylated SCP1 is targeted to the plasma membrane and negatively regulates angiogenesis

Peng Liao[1,2†], Weichao Wang[1,2†], Yu Li[2†], Rui Wang[2†], Jiali Jin[2†], Weijuan Pang[2], Yunfei Chen[1], Mingyue Shen[2], Xinbo Wang[2], Dongyang Jiang[3], Jinjiang Pang[3], Mingyao Liu[2], Xia Lin[4], Xin-Hua Feng[5], Ping Wang[1,2,6*], Xin Ge[7*]

[1]Department of Central Laboratory, Shanghai Tenth People's Hospital, Tongji University School of Medicine, Shanghai, China; [2]Institute of Biomedical Sciences and School of Life Sciences, East China Normal University, Shanghai, China; [3]Department of Cardiology, Shanghai Tenth People's Hospital, Tongji University School of Medicine, Shanghai, China; [4]Department of Surgery, Baylor College of Medicine, Houston, United States; [5]Life Sciences Institute and Innovation Center for Cell Signaling Network, Zhejiang University, Zhejiang, China; [6]School of Life Science and Technology, Tongji University, Shanghai, China; [7]Department of Clinical Laboratory Medicine, Shanghai Tenth People's Hospital, Tongji University School of Medicine, Shanghai, China

**Abstract** SCP1 as a nuclear transcriptional regulator acts globally to silence neuronal genes and to affect the dephosphorylation of RNA Pol ll. However, we report the first finding and description of SCP1 as a plasma membrane-localized protein in various cancer cells using EGFP- or other epitope-fused SCP1. Membrane-located SCP1 dephosphorylates AKT at serine 473, leading to the abolishment of serine 473 phosphorylation that results in suppressed angiogenesis and a decreased risk of tumorigenesis. Consistently, we observed increased AKT phosphorylation and angiogenesis followed by enhanced tumorigenesis in *Ctdsp1* (which encodes SCP1) gene - knockout mice. Importantly, we discovered that the membrane localization of SCP1 is crucial for impeding angiogenesis and tumor growth, and this localization depends on palmitoylation of a conserved cysteine motif within its NH2 terminus. Thus, our study discovers a novel mechanism underlying SCP1 shuttling between the plasma membrane and nucleus, which constitutes a unique pathway in transducing AKT signaling that is closely linked to angiogenesis and tumorigenesis.

**\*For correspondence:** wangp@ tongji.edu.cn (PW); xin.ge@tongji. edu.cn (XG)

[†]These authors contributed equally to this work

**Competing interests:** The authors declare that no competing interests exist.

## Introduction

Tumor angiogenesis is required for tumor growth and metastasis since tumors cannot grow without nutrients and oxygen when their diameters are beyond 1–2 mm, which is considered a distinct characteristic of cancer progression from early to terminal stages (*Folkman, 1971*; *Nicholson and Theodorescu, 2004*). Angiogenesis inhibition approaches have emerged as attractive and promising strategies in anti-cancer treatment. AKT is a central protein kinase in various cell activities, especially for angiogenesis, tumor growth, and progression (*Vivanco and Sawyers, 2002*; *Manning and Cantley, 2007*). Therefore, therapeutic strategies targeting the inhibition of AKT activity in cancer treatments have drawn great public attention during recent decades.

AKT is activated by different extracellular signals such as GPCRs (G protein-coupled receptor), growth factors, and integrins (*Hemmings and Restuccia, 2015*, *2012*). The activation of AKT is

eLife digest Cancerous tumors are the leading cause of death worldwide. Tumors cannot grow beyond a couple of millimeters in diameter unless they are supplied with nutrients and oxygen. To receive these, tumors connect to the body's blood supply by stimulating the growth of new blood vessels. Drugs that reduce the ability of new blood vessels to form have therefore been investigated as possible anti-cancer treatments.

New blood vessels emerge from pre-existing ones in a process called angiogenesis. The first stage of angiogenesis involves the endothelial cells that line the inner wall of the blood vessels moving outwards to form new 'sprouts'. Within the endothelial cells, a signaling protein called AKT drives angiogenesis by moving to the cell membrane, where it is activated and triggers further signaling events. The activation of AKT occurs via a phosphate group being attached to a particular site on the protein.

Enzymes called phosphatases remove phosphate groups from proteins and so can inactivate AKT, hence preventing angiogenesis. Although some phosphatases are known to inactivate AKT, they cannot easily be counted or analyzed. This means that they cannot be used to develop new cancer treatments. In addition, for the phosphatase to best prevent tumor growth, it should inactivate AKT at the cell membrane.

Liao, Wang, Li, Wang, Jin et al. now show that a phosphatase called SCP1 can localize to the cell membrane and inactivate AKT there. SCP1 was not previously known to anchor to the cell membrane. Liao et al. found that this anchoring occurs via a modification that attaches a fatty acid molecule to SCP1.

Further experiments showed that mice that lacked SCP1 had increased levels of AKT phosphorylation in their endothelial cells, more new blood vessel growth and, consequently, had tumors that grew faster.

Further research is now needed to investigate exactly how SCP1 moves to the cell membrane from elsewhere in the cell. Ultimately, this knowledge could play an important role in identifying potential drugs that prevent or reduce the growth of tumors.

initiated by its recruitment to the cell membrane through the interaction of its PH (Pleckstrin homology) domain with PIP3 on the plasma membrane (*Franke et al., 1997*). A large body of evidence shows that AKT activation is closely correlated with its phosphorylation at amino acid serine 473 (S473), while full activation of AKT requires its phosphorylation at S473 and threonine 308 (T308) (*Ramaswamy et al., 1999*; *Kawase et al., 2009*). Activated AKT consequently regulates various downstream targets such as GSK3 and the transcription factor Foxos by direct phosphorylation (*Hemmings and Restuccia, 2012*; *Cohen and Frame, 2001*). Interestingly, accumulating data have shown that AKT can be activated through its S473 phosphorylation by a large variety of molecules and relay a stimulus to downstream oncogenic processes (*Ju et al., 2014*; *Morrison Joly et al., 2016*; *Meric-Bernstam et al., 2014*). In addition, recent discoveries implicate that phosphorylated AKT on S473 may function as an important parameter of oncogenesis and cancer therapy. Likewise, AKT was found to relocate to the nucleus of resistant cells, where it was phosphorylated at S473 by DNA-dependent protein kinase, and this activation inhibited cisplatin-mediated apoptosis in cervical cancers (*Stronach et al., 2011*). Furthermore, cholesterol increased AKT S473 phosphorylation, leading to enhanced tumor growth and a greater number of spontaneous metastases to the lungs in $Apoe^{-/-}$ mice, whereas cholesterol depletion in the cell membrane abrogated AKT S473 phosphorylation, suggesting AKT S473 phosphorylation/dephosphorylation may take place at the plasma membrane (*Alikhani et al., 2013*).

Protein phosphorylation is a reversible process that is mediated by kinases and phosphatases. Compartmentalized phosphorylation/dephosphorylation is a key switch for controlling protein activation and inactivation and complex signal transduction in various biological processes. It has been reported that AKT activity is negatively regulated by protein phosphatases. For example, ubiquitously expressed protein phosphatase 1 (PP1) and PP2A can suppress AKT activity by direct dephosphorylation of AKT at T308 (*Ivaska et al., 2002*; *Resjö et al., 2002*; *Yellaturu et al., 2002*; *Xu et al.,*

*2003*). PH domain-containing proteins such as the phosphatases PHLPP1 and PHLPP2 are able to dephosphorylate AKT at both S473 and T308 (the other site required for full AKT activation) in tumors (*Brognard et al., 2007*). Unfortunately, so far, the known phosphatases involved in AKT inhibition are ubiquitously expressed in cells, which make those phosphatases difficult to be quantified and analyzed under living conditions. Therefore, these phosphatases are not applicable as observational tools for screening potential drugs and novel therapeutic targets in cancer treatments. Moreover, intracellular AKT activity is actually the cause of compromised signals that are orchestrated by various signaling pathways and imbalanced with the direct input signals of drug loading. In pursuit of optimal outcomes of cancer therapies, it is important to identify a unique membrane-localized phosphatase for AKT S473 inactivation, which is the upstream target relaying AKT signals directly from outside the drug stimulus and is the prerequisite of blocking AKT oncogenic signaling from the initial step.

Small CTD phosphatases (SCPs) belong to a family of metal-dependent serine/threonine phosphatases containing a Mg$^{2+}$ binding DXDX (T/V) motif (*Kamenski et al., 2004*). SCPs were originally identified as small CTD containing Pol II phosphatases that shared a similar phosphatase domain with the Pol II CTD phosphatase FCP1 (*Yeo et al., 2003*). SCPs are evolutionarily conserved transcriptional co-repressors for silencing neuronal gene expression via their interaction with the REST/NRSF (RE1 Silencing Transcription Factor or Neural Restrictive Silencing Factor) complex (*Yeo et al., 2005*). SCPs, identified as SCP1 (protein-encoding gene *Ctdsp1*), SCP2, and SCP3, are involved in the TGF-$\beta$ pathway and dephosphorylate distinct oncoproteins such as Snail, promyelocytic leukemia, and c-Myc (*Knockaert et al., 2006*; *Wrighton et al., 2006*; *Wu et al., 2009*; *Lin et al., 2014*). SCP1 and SCP2 are found to be localized in the nucleus, whereas SCP3 is found in both the cytosol and intracellular membranes (*Yeo et al., 2005*; *Wu et al., 2009*; *Siniossoglou et al., 2000*; *Visvanathan et al., 2007*). Despite a large body of evidence indicating that the proper subcellular location of signal molecules is crucial for the accurate signal transduction that accounts for normal biological functions (*Hung and Link, 2011*), the mechanisms underlying the trafficking diversion of SCPs are still unknown.

Protein palmitoylation through the thioester linkage of 16-carbon fatty acids to cysteine residues is unique in that it is the only reversible lipid modification (*Bijlmakers and Marsh, 2003*; *Smotrys and Linder, 2004*; *Aicart-Ramos et al., 2011*). Palmitoylation governs protein function in many ways: palmitoylation can promote membrane translocation of protein and it can facilitate vesicle trafficking and thus contribute to cellular signaling transduction. In neurons and cardiomyocytes, palmitoylation plays an important role in organ development, synaptic plasticity, and the establishment of membrane excitation platforms by clustering various ion channels and transporters (*Kang et al., 2008*; *Fukata and Fukata, 2010*; *Fujiwara et al., 2016*). Moreover, palmitoylation is intimately involved in signaling networks by interacting with various protein molecules such as insulin in the control of phenotypic and functional changes of endothelial cells (*Wei et al., 2014*). A previous study has shown that SCP1 is a palmitoylation substrate by mass spectrometry, whereas direct evidence to prove SCP1 palmitoylation is missing (*Martin and Cravatt, 2009*).

In this study, we find that SCP1, a known nuclear phosphatase, can dephosphorylate AKT by screening a phosphatase library to search for a potential phosphatase that is capable of inactivating AKT. To our surprise, SCP1 is found to be largely located at the plasma membrane by tracing its cellular localization using EGFP-fused SCP1 and SCP1 proteins with different epitope tags. Such membrane-bound SCP1 specifically dephosphorylates AKT at S473 and suppresses angiogenesis, thereby decreasing tumorigenic risk and subsequent tumor growth of lung carcinoma cell-inoculated nude mice. In parallel, increased AKT phosphorylation and promoted angiogenesis, together with a notable risk of tumorigenesis, were observed in SCP1-knockout mice. Moreover, the membrane localization of SCP1 is majorly dependent on the palmitoylation of a conserved cysteine motif within its NH$_2$ terminus, which has a prominent role in SCP1 shuttling between the plasma membrane and nucleus, and thus halting angiogenesis and tumor growth. Collectively, our findings reveal the distinct role of SCP1 as a palmitoylation-dependent phosphatase that negative regulates AKT-mediated angiogenesis and tumorigenesis.

## Results

### SCP1 is localized to the plasma membrane

To explore the functional role of SCP1 in the inhibition of AKT activity, we surprisingly found that both GFP-SCP1 and Flag-SCP1 were mainly localized at the plasma membrane in HeLa cells, where they were well co-localized with the plasma membrane marker GFP-PLCδ-PH (*Figure 1A*). Similar results were also obtained in various cell lines, including MDCK, COS7, MCF7, HEK293T, and DLD1 cells (*Figure 1—figure supplement 1A*). To rule out the artificial intervention from the fused GFP protein, we compared the subcellular localizations of untagged and GFP-fused SCP1 at the N terminus or C terminus, respectively. Our data demonstrated that both untagged and GFP-fused SCP1 showed similar membrane localizations (*Figure 1—figure supplement 1B*), suggesting that SCP1 is a membrane-localized protein. SCP2 and SCP3, which are structurally similar to SCP1, appeared to possess an identical distribution pattern at the plasma membrane when they were transiently expressed in 293 T cells, whereas CTDSPL2 (SCP4) was mainly localized in the nucleus (*Figure 1—figure supplement 1C*). Cell fractionation experiments confirmed that most part of the SCP1 was in the membrane fraction, with a small proportion in the nuclear fraction, but none of the SCP1 was found in the cytosolic fraction (*Figure 1B*). As expected, endogenous SCP1 was also easily detected at the plasma membrane using cell membrane fraction assays (*Figure 1C*). The membrane localization of SCP1 was further confirmed using FRAP (The Ferric Reducing Ability of Plasma) assays and time-lapse microscopy (*Figure 1—figure supplement 1D*). Interestingly, treatment with brefeldin A (BFA), a blocker of protein trafficking from the Golgi to the plasma membrane, had little effect on the membrane localization of SCP1 (*Figure 1—figure supplement 1E*), suggesting that the membrane-localized SCP1 proteins are not newly synthesized through Golgi/endosome trafficking. Taken together, our data clearly demonstrate that SCP1 is a plasma membrane-associated phosphatase.

To further dissect the molecular basis of the membrane-dependent localization of SCP1, serial deletion mutants (*Figure 1D*) were generated and their subcellular localizations were detected by immunostaining. As shown in *Figure 1E*, the SCP1 mutants containing the residues from 31 to 55 were able to target GFP to the plasma membrane, suggesting that the protein region of SCP1 from residues 31 to 55 is essential for targeting SCP1 to the plasma membrane.

### Palmitoylation of SCP1

We next determined the underlying mechanism of SCP1 membrane localization. Because we could not find any transmembrane region within SCP1, we examined whether its membrane localization is mediated by lipid modifications. To this end, we examined the effects of various lipid modification inhibitors, including the farnesyltransferase inhibitor FTI-227, the prenyl transferase inhibitor GGTI, and the palmitoylation inhibitor 2-bromopalmitate (2BP) on SCP1 membrane localization (*Rowinsky et al., 1999*). We did not detect any effect of FTI-227 and GGTI on SCP1 membrane localization (*Figure 2—figure supplement 1A*). However, FTI-227 and GGTI treatments dramatically blocked the membrane localization of H-Ras in control experiments (*Figure 2—figure supplement 1A*). In sharp contrast, treatment with 2BP for 4 h resulted in a remarkable reduction of membrane-localized SCP1 and significantly increased cytosolic and nuclear SCP1 (*Figure 2A* and *Figure 2—figure supplement 1B*). This result was confirmed by cell fractionation assay, which showed that the membrane-associated SCP1 level was markedly reduced upon 2BP treatment. In parallel, the cytosolic and nuclear distribution of SCP1 increased (*Figure 2B*). Similar results were obtained from SCP2- and SCP3-transfected cells (*Figure 2—figure supplement 1C*). Interestingly, a small amount of SCP1 was also found on the Golgi membrane after 8 h of 2BP treatment (*Figure 2—figure supplement 1D*). These data indicate that membrane recruitment of SCP1 is mediated by palmitoylation, but not farnesylation or prenylation, which offers direct evidence for the previous report that identified SCP1 as a palmitoylation substrate (*Martin and Cravatt, 2009*).

It has been reported that palmitoylated proteins can be recycled from the plasma membrane to the Golgi (*Resh, 2006*). Therefore, we tested whether the nucleus- or Golgi-localized SCP1 was newly synthesized and recycled to the nucleus or Golgi from the plasma membrane. SCP1 localization was monitored in transfected HeLa cells treated with cycloheximide (CHX) for 8 h to block new protein synthesis in the presence or absence of 2BP, which demonstrated that CHX had little effect on the membrane localization of SCP1 (*Figure 2—figure supplement 1D*). However, in the presence

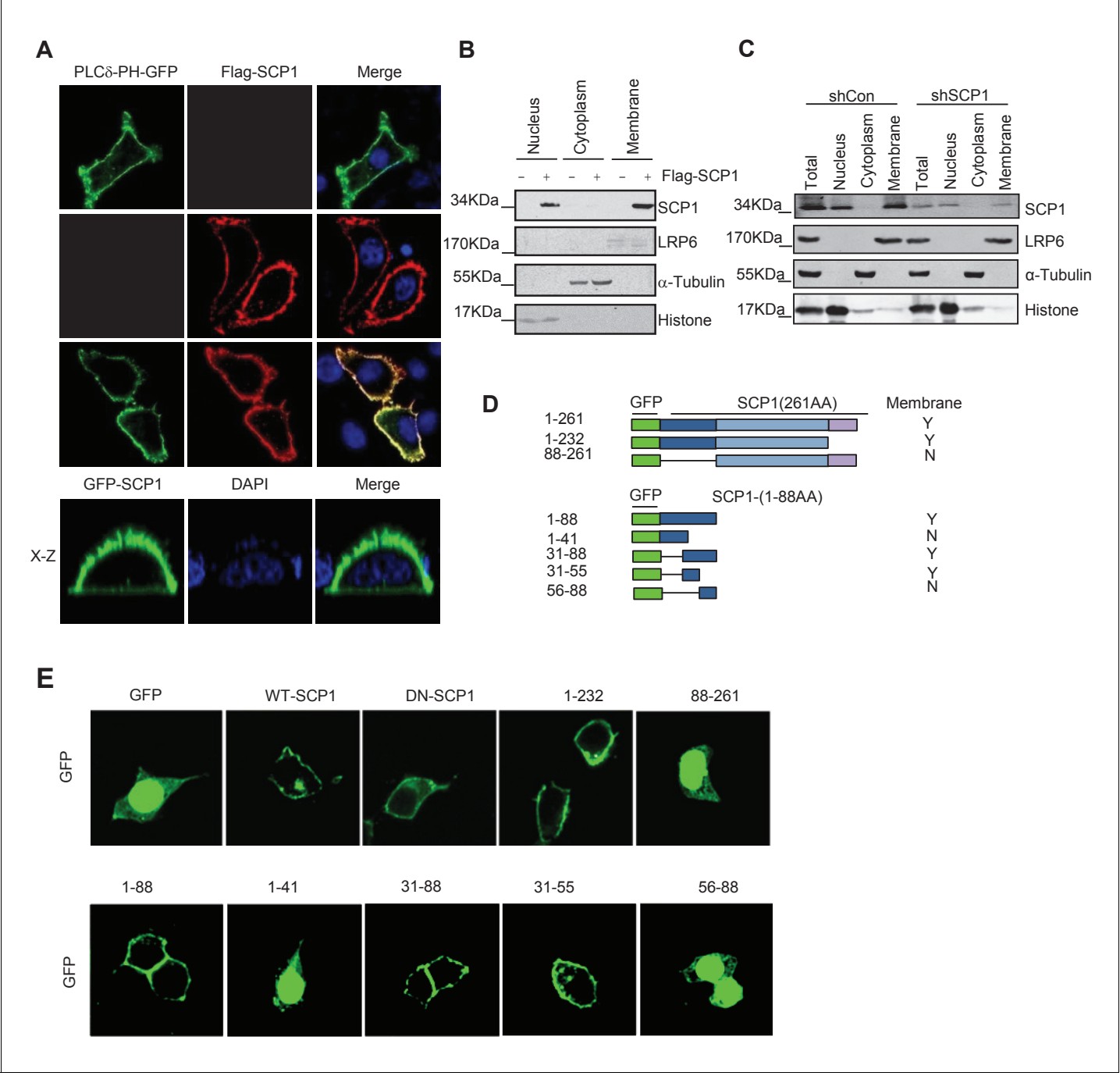

**Figure 1.** Membrane localization of SCP1 (A) SCP1 was co-localized with PLCδ-PH on the cell membrane. FLAG-SCP1 was transfected with or without PLCδ-PH-GFP in HeLa. The subcellular localization of SCP1 and PLCδ-PH-GFP was analyzed using immunofluorescence assay, and both the horizontal section (X–Y) and vertical section (X–Z) were photographed. (B) and (C) Subcellular localization of SCP1 in cells. HEK-293T cells were transfected and the subcellular localizations of transfected SCP1 (B) or endogenous SCP1 (C) were analyzed using western blotting. (D) Cartoon of different deletion mutations of SCP1. Yes (Y) and no (N) represent SCP1 or truncated mutant membrane localizations, respectively. (E) HeLa cells were transfected with GFP-SCP1 or its mutants for 24 h and then analyzed for their subcellular localization using immunofluorescence assays.

The following figure supplement is available for figure 1:

**Figure supplement 1.** SCP1 is membrane localized.

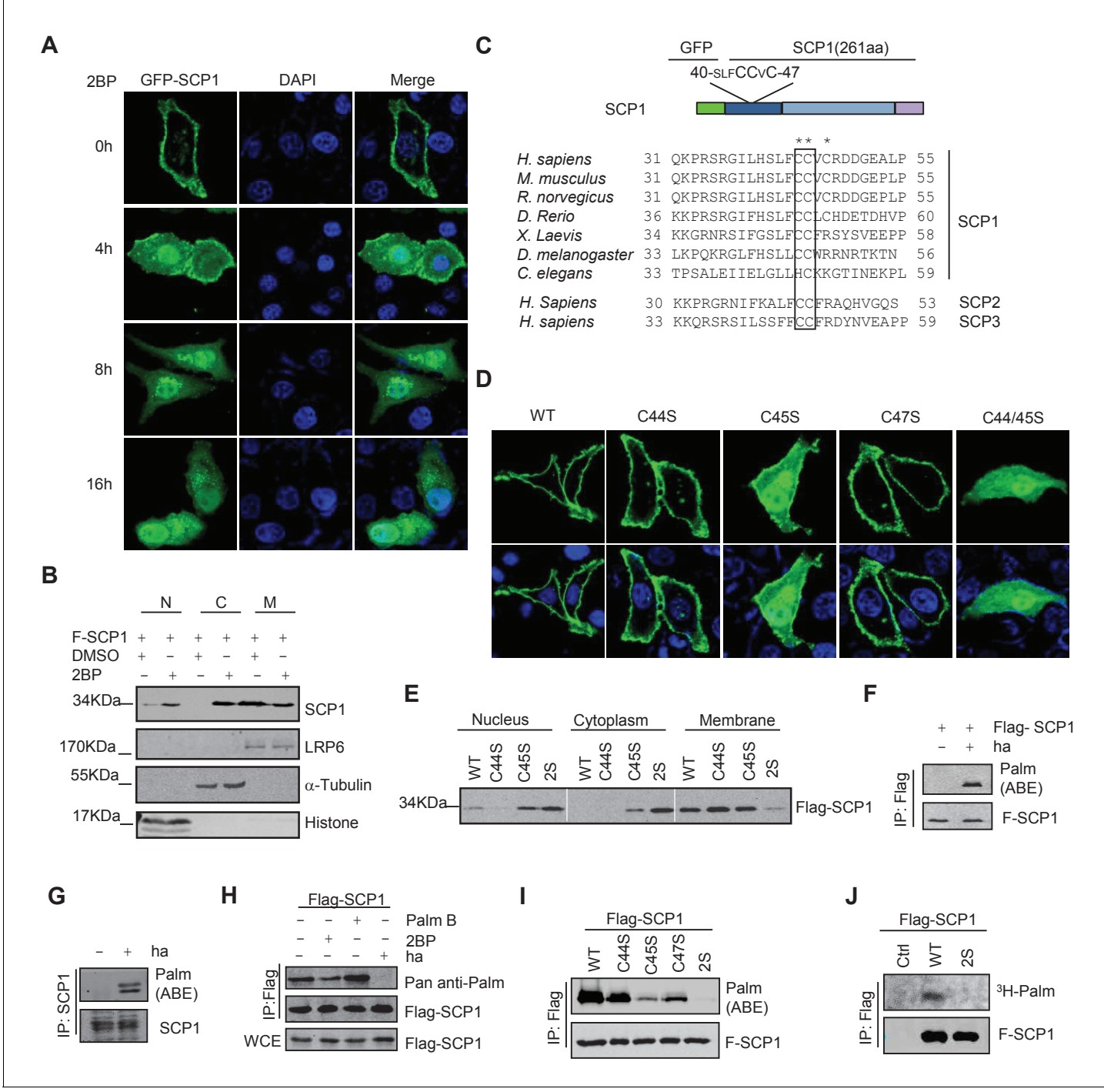

**Figure 2.** SCP1 was palmitoylated (A) Palmitoylation inhibitor 2-bromopalmitate (2BP) blocked the SCP1 membrane localization. HeLa cells were transfected and treated with 2BP (10 µM) for 4, 8, or 16 h or DMSO as a control. The subcellular localization of SCP1 was detected using immunofluorescence assay. (B) HEK293T cells were treated with DMSO or 2BP (10 µM) for 6 h and the subcellular location of SCP1 was detected using western blotting. (C) Potential palmitolylation sites Cys44 and Cys45 of SCP1 were evolutionarily conserved. Amino acids 33–55 of SCP1 in different species, ranging from *Caenorhabditis elegans* to *Homo sapiens*, 30–53 of SCP2 in *H. sapiens*, and 33–59 of SCP3 in *H. sapiens* are shown. (D) HeLa cells were transfected with WT-SCP1, C44S-SCP1, C45S-SCP1, C47S-SCP1, and C44/45S(2S)-SCP1 for 24 h. The subcellular localizations of WT-SCP1 and its mutants were detected using immunofluorescence assay. (E) HEK293T cells were transfected with WT-SCP1, C44S-SCP1, C45S-SCP1, and C44/45S(2S)-SCP1 for 24 h and cell fractions of were analyzed using western blotting. (F) FLAG-SCP1 was expressed in HEK293T cells, immunoprecipitated, and palmitoylation was detected using the acyl–biotin exchange (ABE) assay. (G) Palmitoylation of endogenous SCP1 in HEK293T cells was detected using the ABE assay. (H) FLAG-SCP1 was expressed in HEK293T cells for 24 h and treated with 2BP (10 µM) or palmostatin B (50 µM) for 12 h. Palmitoylation

*Figure 2 continued on next page*

*Figure 2 continued*

of SCP1 was detected using pan-palmitoylation antibody. (**I**) and (**J**) Identification of palmitoylation sites using the ABE assay (**I**) and the [³H] palmitate incorporation assay (**J**).

The following figure supplement is available for figure 2:

**Figure supplement 1.** SCP1 membrane localization depends on its palmitoylation.

of 2BP, CHX treatment significantly reduced the distribution of SCP1 on the Golgi membrane, but minimally affected its nuclear accumulation (*Figure 2—figure supplement 1D*). These data indicate that the newly synthesized SCP1 protein is translocated to the Golgi without being palmitoylated and then targeted to the plasma membrane by palmitoylation. The pool of SCP1 at the plasma membrane was constitutively depalmitoylated and the 2BP treatment prevented its re-palmitoylation, leading to the redistribution of depalmitoylated SCP1 in cytosolic and nuclear compartments (*Figure 2—figure supplement 1E*). These data suggest that palmitoylation is required for SCP1 localization at the plasma membrane, while depalmitoylation allows SCP1 to be recycled from the cell surface to the nucleus.

As mentioned above, this is a reversible modification as palmitoylation involves the attachment of a thioester chain to cysteine residues, and this controls the transient membrane targeting of peripheral membrane proteins (so-called S-palmitoylation) (*el-Husseini et al., 2002*). We found that SCP1 contains an evolutionarily conserved di-cysteine motif at its membrane-associated region, which constitutes a potential palmitoylation site predicted by ExPASy or CSS-Palm software (*Figure 2C*). The membrane-localized N-terminal fragment of SCP1 containing putative S-palmitoylation sites is sensitive to 2BP treatment (*Figure 2—figure supplement 1F*). Therefore, we examined the functional role of these conserved cysteines for targeting SCP1 to the membrane by mutating these cysteines into serines. As shown in *Figure 2D*, the individual mutations of Cys44 (C44), Cys45 (C45), or Cys47 (C47) to Ser had minimal effects on the membrane distribution of SCP1. However, the simultaneous mutation of both Cys44 and Cys45 into serines (2S) resulted in a dramatic reduction of membrane-associated SCP1 by 90% (*Figure 2D*). The C45S and C44/45S mutations also significantly increased the levels of SCP1 in the cytosol and nucleus (*Figure 2E*). These data indicate that SCP1 binds to the plasma membrane through the conserved cysteines of Cys44 and Cys45.

To test whether SCP1 was palmitoylated in vivo, a cysteine accessibility assay (also named the acyl–biotin exchange [ABE] assay) was used (*Noritake et al., 2009*). We found that both overexpressed and endogenous SCP1 were palmitoylated in HEK293T cells (*Figure 2F and G*). It is well established that a distinguishing feature of S-palmitoylation is its reversibility. Since both the forward and reverse reactions take place in a reversible fashion, we also validated the dynamic palmitoylation of SCP1 using both the palmitoylation inhibitor (2BP) and the depalmitoylation inhibitor (palmostatin B) (*Dekker et al., 2010*). As expected, the palmitoylation of SCP1 can be regulated by both 2BP and palmostatin B in an opposing fashion, as shown by the decreased palmitoylation of SCP1 by 2BP versus the increased palmitoylation of SCP1 by palmostatin B via using pan Anti-palmitoylation Antibody (*Fang et al., 2016*) (*Figure 2H*), suggesting a dynamic balance of SCP1 inside and outside of the plasma membrane. The residues required for palmitoylation were also examined. As shown in *Figure 2I*, all of the C44S, C45S, and C47S mutations led to a reduction of palmitoylation to different extents, while the SCP1 C44/45S double mutant almost completely abolished palmitoylation. The palmitoylation level of the C45S mutant was much less than that of the C44S or C47S mutants, which matched these mutants to their role in SCP1 membrane association. The palmitoylation of wild-type (WT) SCP1 was also confirmed by a [³H] palmitic acid incorporation assay. The 2S mutant showed a remarkably reduced level of [³H] palmitic acid incorporation (*Figure 2J*), suggesting that such cysteines are the major palmitoylation sites of SCP1. Taken together, these results indicate that SCP1 is palmitoylated at its N-terminus, which accounts for the targeting SCP1 of to the plasma membrane.

## SCP1 is a negative regulator of angiogenesis

To further clarify the function of SCP1, we generated SCP1-knockout (KO) mice (*Figure 3—figure supplement 1A and B*). The KO efficiency of the *Ctdsp1* gene was confirmed by gene sequencing, polymerase chain reaction (PCR), and western blotting (*Figure 3—figure supplement 1C–E*). The SCP1-KO mice gave birth at a typical median ratio and developed normally (data not shown). Because palmitoylation plays different roles in neurogenesis and is tightly controlled by angiogenesis in that growth factors such as VEGF perform multiple functions (*Sun et al., 2003*; *Lee et al., 2007*), we examined whether SCP1 is involved in embryonic vasculogenesis using SCP1-KO mice. To this end, we analyzed in vivo retinal vasculogenesis at postnatal day 5 (P5) using whole-mount and imaging techniques. We found that the vascular areas were larger in the SCP1-KO mice than those of WT mice (*Figure 3A*). We also found significantly increased vascular sprouts and branch points in SCP1-KO mice (*Figure 3B*). We further examined the effect of SCP1 on postnatal angiogenesis using a hind limb ischemia model and found a significantly promoted hind limb flow recovery in SCP1-KO mice (*Figure 3C*). Because angiogenesis is essential for tumorigenesis (*Testa and Bellacosa, 2001*), we also explored the role of SCP1 on tumor growth by injecting Lewis lung carcinoma cells (LLCs) into WT or SCP1-KO mice (*O'Reilly et al., 1994*). Our data showed that SCP1 KO significantly promoted tumor growth (*Figure 3D* and *Figure 3—figure supplement 1F*). Accordingly, we found that SCP1 deficiency significantly enhanced angiogenesis in tumors (*Figure 3E*). Thus, our data indicate that SCP1 is a negative regulator of angiogenesis.

AKT is intimately involved in the regulation of angiogenesis. VEGF can regulate angiogenesis through the PI3K/AKT pathway (*Shiojima and Walsh, 2002*). Next, we examined whether the effect of SCP1 on angiogenesis depends on AKT activation. To this end, we performed an aortic ring assay by in vitro culturing thoracic aorta from WT or SCP1-KO littermates (*Kitamura et al., 2008*). We found that the aortic rings from SCP1-KO mice showed an increased capability of angiogenesis, evidenced by enlarged areas of capillary sprouting (*Figure 3F*). MK2206 was reported as an allosteric AKT inhibitor that enhanced the antitumor efficacy of standard chemotherapeutic agents in vitro and in vivo (*Hirai et al., 2010*). Treatment with MK2206 reversed the above-mentioned effects in SCP1-KO mice (*Figure 3F*). These results indicate that SCP1 negatively regulates angiogenesis in an AKT-dependent manner.

Because the endothelial cell is the major cell type involved in angiogenesis, we investigated whether SCP1 deficiency impairs the function of endothelial cells by examining the functional role of SCP1 in HUVECs (a human endothelial cell line from umbilical veins). To this end, we first examined whether SCP1 regulates AKT phosphorylation in HUVECs. Our data showed that SCP1 depletion significantly promoted VEGF-induced AKT activation in HUVECs (*Figure 4A*). Moreover, we found that SCP1 overexpression markedly inhibited tubule formation and cell migration (*Figure 4B* and *Figure 4—figure supplement 1A*). Importantly, the inhibitory effect of SCP1 was rescued by the coexpression of the AKT-S473D mutant (*Figure 4B* and *Figure 4—figure supplement 1A*). In parallel, SCP1 depletion markedly increased angiogenesis and endothelial cell migration, which could be reversed by AKT inhibitors (*Figure 4C and D*). These data indicate that SCP1 suppresses angiogenesis in an AKT phosphorylation-dependent fashion.

## SCP1 is a novel phosphatase that dephosphorylates AKT

To test whether SCP1 can directly regulate AKT, we expressed a constitutively active form of AKT, herein referred as Myr-AKT, in which the N-terminus is fused with a myristoylation signal and localized at the plasma membrane (*del Peso et al., 1997*). We co-expressed SCP1 with Myr-AKT in HEK293T cells and found that the co-expression of SCP1, but not its catalytically inactive mutant DN-SCP1 (in which Asp96 is substituted to Glu) (*Wrighton et al., 2006*), reduced the phosphorylation level of Myr-AKT at Ser473 by 80% (*Figure 5A*). To examine whether SCP1 can directly dephosphorylate AKT, Myr-AKT was incubated with GST, GST-WT-SCP1, or GST-DN-SCP1 purified from *Escherichia coli*. We found that WT-SCP1, but not DN-SCP1, markedly dephosphorylated AKT at Ser473, accompanied by a lesser extent of AKT dephosphorylation at Thr308 (*Figure 5B*).

We next examined whether SCP1 overexpression could result in AKT dephosphorylation by immunostaining assay. We found that overexpression of SCP1 significantly diminished the phosphorylation of endogenous AKT at Ser473 without affecting total AKT levels (*Figure 5C*). In addition, SCP1 knockdown by shRNA markedly increased AKT phosphorylation at Ser473 (*Figure 5D*). AKT

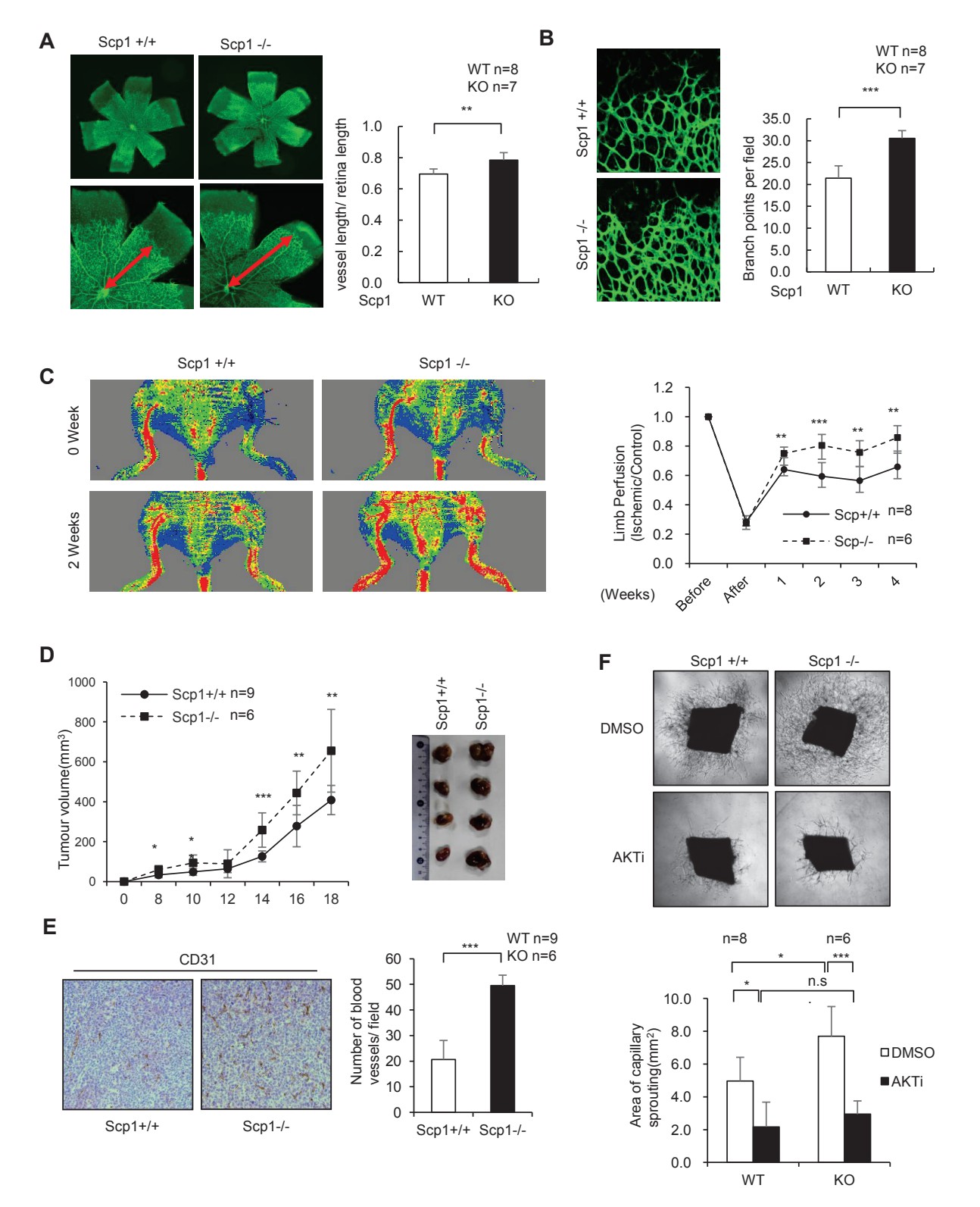

**Figure 3.** SCP1 knockout promoted angiogenesis (**A**) SCP1 deletion impaired the development of retinal angiogenesis. Retinas of postnatal day 5 were isolated from littermates of wild-type (WT; n = 8) or SCP1-knockout (KO) mice (n = 7) and stained with Isolectin B4. Quantification of vessel length was measured, and the rate of vessel length/retina length was calculated. **p<0.01. (**B**) Loss of SCP1 reduced the branching of the vessels. The branching of vessels was counted. ***p<0.001. (**C**) SCP1 deficiency promoted the recovery of hind limb ischemia. The laser Doppler blood flowmetry ratio was

*Figure 3 continued on next page*

*Figure 3 continued*

significantly higher in SCP1-KO mice (n = 6) than in WT mice (n = 8). ***p<0.001, **p<0.01. (D) SCP1 KO promoted Lewis lung carcinoma cell (LLC) tumor bearing in C57 mouse. 2 × 10⁵ LLC cells were injected into SCP1-WT or -knockdown littermates. The diameter of the tumor was measured every 2 days, and the volume of the tumor was calculated. ***p<0.001, **p<0.01, *p<0.05. (E) Angiogenesis was promoted in SCP1-KO mice. The angiogenesis in tumors was analyzed using immunohistochemistry by CD31 staining. (F) SCP1 deficiency promoted angiogenesis in an AKT-dependent manner. Segments (1 mm in length) of the aorta from SCP1-WT (n = 8) or SCP1-KO (n = 6) mice were embedded in Matrigel and treated with DMSO or AKT inhibitor (MK2206, 2 nM) for 6 days. Sprouting was observed and photographed by microscopy. The vascular area of each group was measured using Image J. ***p<0.001, **p<0.01, *p<0.05.

The following source data and figure supplement are available for figure 3:

**Source data 1.** SCP1 knockout promoted angiogenesis
**Figure supplement 1.** Generation and validation of SCP1-knockout mice.

can be activated by various upstream signals, such as growth factors, hormones, and cytokines (*Rodon et al., 2013*). Therefore, we tested whether SCP1 could block AKT activation under such conditions. We found that EGF treatment markedly induced the membrane distribution of AKT and AKT phosphorylation at Ser473 in HeLa cells. In addition, SCP1 expression suppressed EGF-induced AKT phosphorylation at Ser473 without affecting AKT distribution at the plasma membrane (*Figure 5—figure supplement 1A*). Meanwhile, both EGF- and insulin-induced AKT phosphorylation at Ser473 in H1299 cells can be significantly elevated by SCP1 depletion (*Figure 5E* and *Figure 5—figure supplement 1B*). The knockdown efficiency of SCP1 was measured at the mRNA and protein level (*Figure 5—figure supplement 1C*). We also found that SCP1 depletion significantly enhanced the expression levels of AKT-phosphate substrates as examined by anti-AKT-phosphate substrate antibody (*Figure 5E*). Similar results were obtained using MEF cells derived from SCP1-WT and -KO littermates under the insulin treatment (*Figure 5F*). Phosphatase PHLPP has been implicated in cleaving pS473 from AKT (*Cailliau et al., 2015*; *Qiao et al., 2007*). We found that SCP1 or PHLPP depletion could promote AKT phosphorylation at Ser473. Furthermore, double knockdown of SCP1 and PHLPP could exaggerate this effect (*Figure 5—figure supplement 1D and E*). We also overexpressed PHLPP1 when SCP1 was disabled and found that PHLPP1 can still decrease AKT phosphorylation at Ser473 (*Figure 5—figure supplement 1F*). Based on these data, we conclude that the phosphatase function of SCP1 that dephosphorylates AKT is independent of PHLPP activity. Taken together, our results indicate that SCP1 is a phosphatase that dephosphorylates AKT at Ser473.

The phosphorylation of Ser473 is critical for AKT activation. Therefore, we examined whether SCP1 could regulate AKT kinase activity using an in vitro kinase assay. We found that SCP1 overexpression markedly reduced AKT activity (*Figure 5G*). Thus, our data indicate that dephosphorylation of AKT by SCP1 significantly inhibits AKT kinase activity.

Next, we investigated whether SCP1 could interact with AKT in vivo. To this end, SCP1 was immunoprecipitated from HEK293T cells and the association of endogenous AKT and SCP1 was detected by immunoblotting. We found that endogenous AKT could be co-immunoprecipitated by both overexpressed and endogenous SCP1 (*Figure 5H* and *Figure 5—figure supplement 1G*), indicating that SCP1 associates with AKT in cells. In addition, we found that membrane-localized Myr-AKT had a stronger affinity to SCP1 than WT AKT (*Figure 5I*). The interaction between SCP1 and AKT was independent of SCP1 phosphatase activity, since dominant-negative SCP1 still showed an association with AKT (*Figure 5I*). Consistently, AKT could directly bind to the purified SCP1 in vitro, while mutations of AKT at Thr308, Thr450, or Ser473 had little effect on their interaction, suggesting that the direct interaction between SCP1 and AKT is independent of AKT phosphorylation (*Figure 5J* and *Figure 5—figure supplement 1H*).

## Palmitoylation of SCP1 is required for AKT inhibition

Then, we examined whether SCP1 palmitoylation is involved in its inhibitory effect on AKT signaling. As shown in *Figure 6A*, the mutant (SCP1 2S) that cannot be palmitoylated significantly reduced the capacity for dephosphorylating AKT at Ser473 in cells. Accordingly, the 2S mutant abolished the inhibitory capability of SCP1 on cell growth (*Figure 6B* and *Figure 6—figure supplement 1A*).

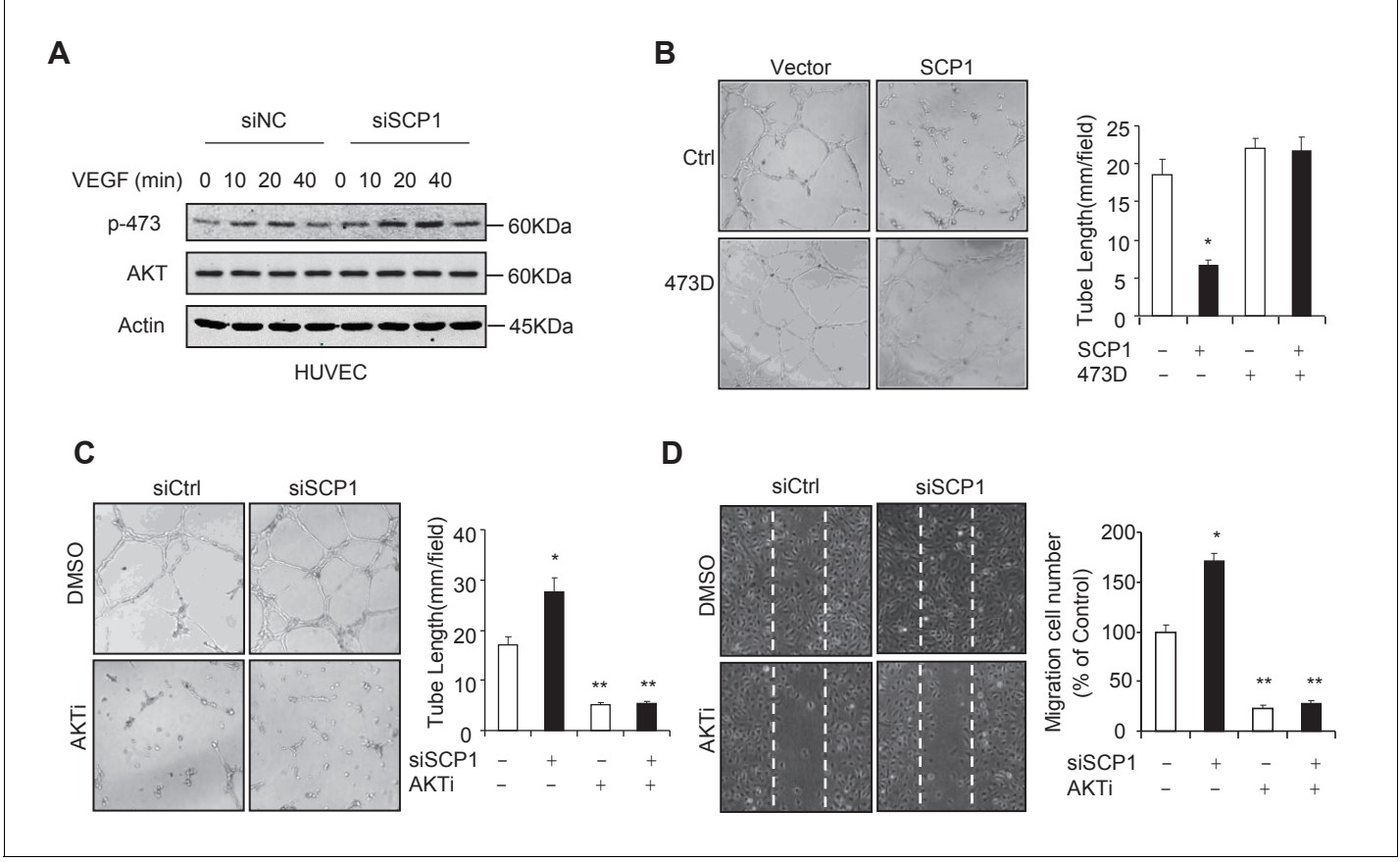

**Figure 4.** SCP1 inhibited AKT-mediated angiogenesis (**A**) SCP1 deletion promoted VEGF-induced AKT activation in HUVECs. HUVECs were transfected with siNC (Small Interfering RNA for Normal Control) or siSCP1 (Small Interfering RNA for SCP1) for 72 h and stimulated with VEGF (100 ng/ml) as indicated after starvation for 8 h. (**B**) SCP1 impaired HUVEC tube formation through AKT. HUVECs were overexpressed with SCP1 with or without AKT-S473D. The cells were placed in plates coated with Matrigel and tubular structures were photographed after 6 h. The tube lengths were measured in each field. *p<0.05. (**C**) SCP1 depletion inhibited the tube formation of HUVECs through AKT. HUVECs were transfected with siSCP1 and treated with or without AKT inhibitor (AKTi; MK2206, 2 nM) for 5 days as indicated. The tube lengths were measured in each field. **p<0.01, *p<0.05. (**D**) SCP1 deletion promoted HUVEC migration through AKT. Cell migration was detected using a wound healing assay. HUVECs were transfected and treated with or without AKTi (MK2206, 2 nM). The migration cell number in each field was calculated. **p<0.01, *p<0.05.

The following source data and figure supplement are available for figure 4:

**Source data 1.** SCP1 inhibited AKT-mediated angiogenesis

**Figure supplement 1.** SCP1 inhibits HUVEC migration.

Meanwhile, 2BP treatment inhibited the ability of SCP1 to dephosphorylate AKT and increased the endogenous level of phosphorylated AKT Ser473 (*Figure 6C*). These data indicate that palmitoylation is pivotal for the suppressive effect of SCP1 on cell growth.

Next, we examined whether palmitoylation is required for the phosphatase activity of SCP1. To this end, HA-tagged WT-SCP1, DN-SCP1, and 2S-SCP1 mutants were expressed, respectively, in HEK293T cells and immunoprecipitated with an anti-HA antibody. The phosphatase activities of the above-mentioned mutants were measured by a pNPP phosphatase (p-nitrophenyl-phosphate) assay (*Wang et al., 2016*). We found that mutations at SCP1 palmitoylation sites did not affect the phosphatase activity of SCP1 (*Figure 6—figure supplement 1B*).

The effect of palmitoylation on the interaction between SCP1 and AKT was also examined. As shown in *Figure 6D*, both treatment with 2BP and mutations at palmitoylation sites (F-SCP1-2S) decreased the interaction between SCP1 and the constitutively active form of AKT (Myr-AKT-HA).

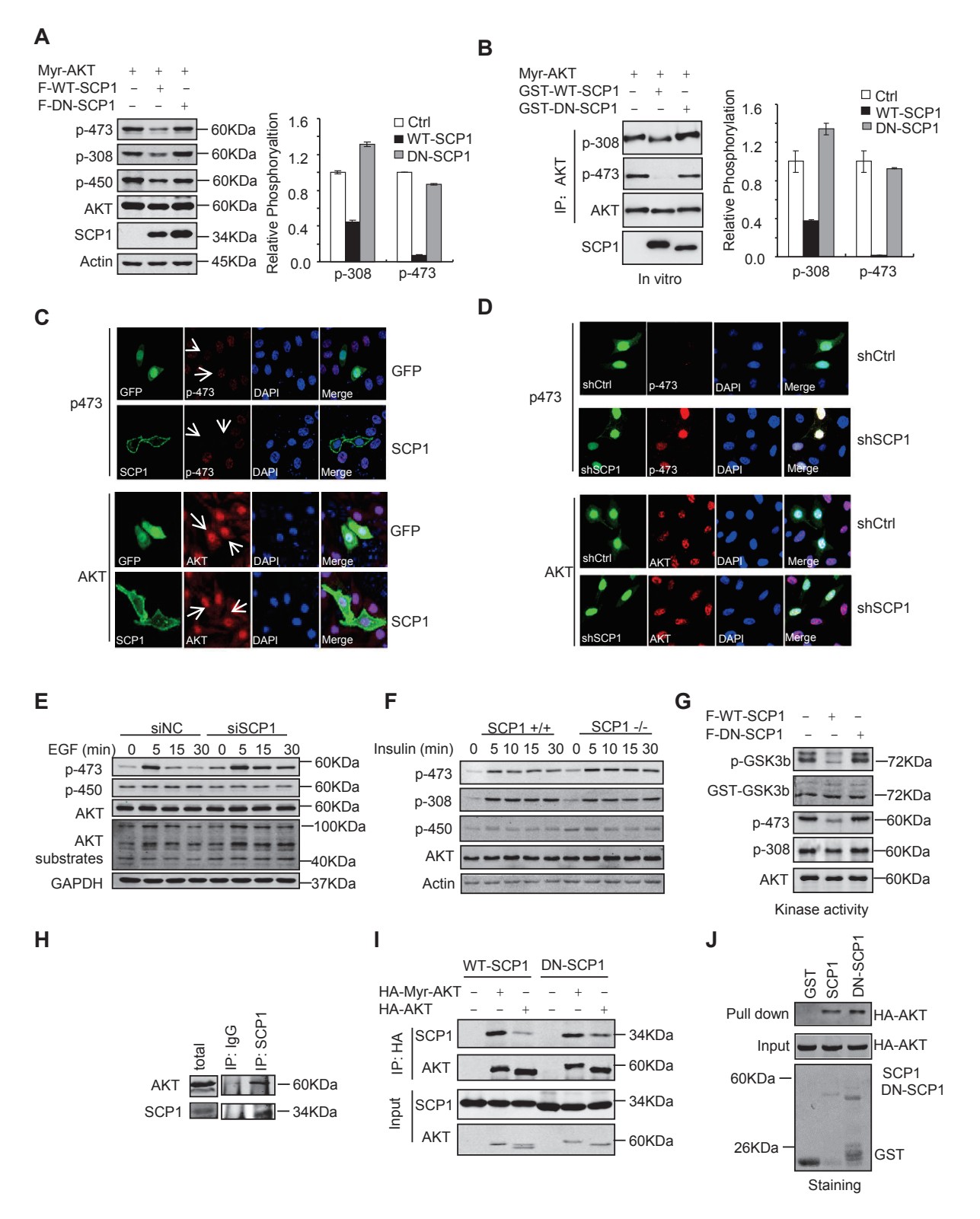

**Figure 5.** SCP1 dephosphorylated AKT (**A**) Wild-type (WT)-SCP1 dephosphorylated AKT Ser473. Myr-AKT was co-expressed with vector, WT-SCP1, or DN-SCP1. The phosphorylations of p-Ser473-AKT, p-Thr308-AKT, and p-Thr450-AKT was analyzed using western blotting. The relative phosphorylations of p-Thr308-AKT and p-Thr450-AKT are displayed in the form of a histogram. (**B**) WT-SCP1 dephosphorylated AKT in vitro. HA-Myr-AKT was immunoprecipitated from HEK293T cells and incubated with purified GST, GST-WT-SCP1, or GST-DN-SCP1 for 30 min. The phosphorylations of

*Figure 5 continued on next page*

Figure 5 continued

p-Ser473-AKT and p-Thr308-AKT were analyzed using western blotting. (C) WT-SCP1 dephosphorylated AKT in HeLa cells. HeLa cells were transfected with GFP-SCP1 for 24 h. The phosphorylation of p-Ser473-AKT and total AKT was detected using immunofluorescence assay. (D) SCP1 knockdown promoted AKT Ser473 phosphorylation in HeLa cells. Control or *Ctdsp1* shRNA was transfected into HeLa cells for 72 h. The phosphorylation of p-Ser473-AKT and total AKT was detected using immunofluorescence assay. (E) SCP1 knockdown promoted EGF-induced AKT activity. H1299 cells were transfected with control or *Ctdsp1* siRNA for 72 h. The cells were stimulated with EGF (100 ng/ml) as indicated after 8 h of starvation, and phosphorylation of AKT was detected using immunofluorescence assay. (F) SCP1 depletion promoted insulin-stimulated AKT activation. *Ctdsp1*$^{+/+}$ or *Ctdsp1*$^{-/-}$MEFs (mouse embryonic fibroblast) were stimulated with insulin (1 mM) as indicated after 6 h of starvation. (G) WT-SCP1 decreased the AKT kinase activity. AKT was transfected into HEK293T cells with vector, WT-SCP1, or DN-SCP1, immunoprecipitated, and incubated with GST-GSK3$\beta$. The phosphorylation of GSK3$\beta$ was measured using western blotting. (H) Endogenous AKT interacted with endogenous SCP1. Endogenous SCP1 was immunoprecipitated using an anti-SCP1 antibody and the associated AKT was detected using an anti-AKT antibody. (I) The interaction of SCP1 with WT or myristoylated AKT1 is independent of its phosphatase activity. (J) Purified GST, GST-WT-SCP1, and GST-DN-SCP1 were incubated with cell lysates overexpressing AKT. The interaction was detected using western blotting.

The following figure supplement is available for figure 5:

**Figure supplement 1.** SCP1 negatively regulates AKT activation on membrane.

Moreover, the mutant 2S-SCP1 was unable to block the migration of HUVECs in a wound healing assay (*Figure 6E* and *Figure 6—figure supplement 1C*), suggesting that palmitoylation of SCP1 affects the functional role of endothelial cells that accounts for angiogenesis. Similar results were obtained for tubule formation using a Matrigel angiogenesis assay (*Figure 6F* and *Figure 6—figure supplement 1D*). We also found that WT-SCP1, but not DN-SCP1 or 2S-SCP1, suppressed tumorigenesis (*Figure 6—figure supplement 1E*). In summary, these data indicate that palmitoylation is essential for SCP1 to dephosphorylate AKT via their interactions, which plays a crucial role in angiogenesis and tumor growth.

## Discussion

SCP1 was initially identified as a nuclear phosphatase that dephosphorylates RNA pol ll (*Yeo et al., 2003*; *Zhang et al., 2006*). In this study, we unexpectedly found that SCP1, SCP2, and SCP3 are mainly localized at the plasma membrane in different cell types in a steady state (*Figure 1* and *Figure 1—figure supplement 1*). Such a novel phenotype observed from SCP1 raises much interest because only a few serine/threonine phosphatases are localized at the plasma membranes, although approximately 30 serine/threonine and 107 tyrosine protein phosphatases have been reported (*Shi, 2009*; *Alonso et al., 2004*). Importantly, all of these membrane-located phosphatases have been demonstrated to play crucial roles in various biological processes. For instance, PTEN can function as a protein phosphatase that localizes at the plasma membrane to dephosphorylate membrane proteins and is intimately involved in tumor progression by inhibiting oncogenic signaling (*Wu et al., 2000*). PH domain-containing family members such as PHLPP1 and PHLPP2 can mediate AKT signaling by recruiting proteins to the plasma membrane through its interaction with PIP2 (*Brognard et al., 2007*). Another study indicated that PHLPP1 interacts with scribble via its PDZ (Post synaptic density protein (PSD95), Drosophila disc large tumor suppressor (Dlg1), and Zonula occludens-1 protein (zo-1)) domain and thereby mediates its localization at the plasma membrane, which negatively regulates AKT-mediated tumorigenic signals (*Hung and Link, 2011*).

In order to explore the mechanisms underlying the membrane distribution of SCP1, we used both in vitro and in vivo animal models and consequently found that SCPs were palmitoylated at a conserved cysteine motif within its N-terminus (*Figure 2* and *Figure 2—figure supplement 1*). S-palmitoylation is a lipid post-translational modification involving the covalent attachment of fatty acid palmitate to cysteine residues via a thioester bond. So far, it is the only reversible lipid modification that has been identified (*Mumby, 1997*). Importantly, palmitoylated proteins were found to play very important roles in multiple biological processes, such as synaptic plasticity during neuronal development (*Fukata and Fukata, 2010*; *El-Husseini et al., 2002*; *Kutzleb et al., 1998*; *Arstikaitis et al., 2008*). In addition, various membrane-localized proteins that are crucial in the maintenance of protein structure and function are palmitoylated (*Rocks et al., 2010*). Unlike other lipid modifications, protein palmitoylation is highly dynamic, and cycles of palmitoylation and

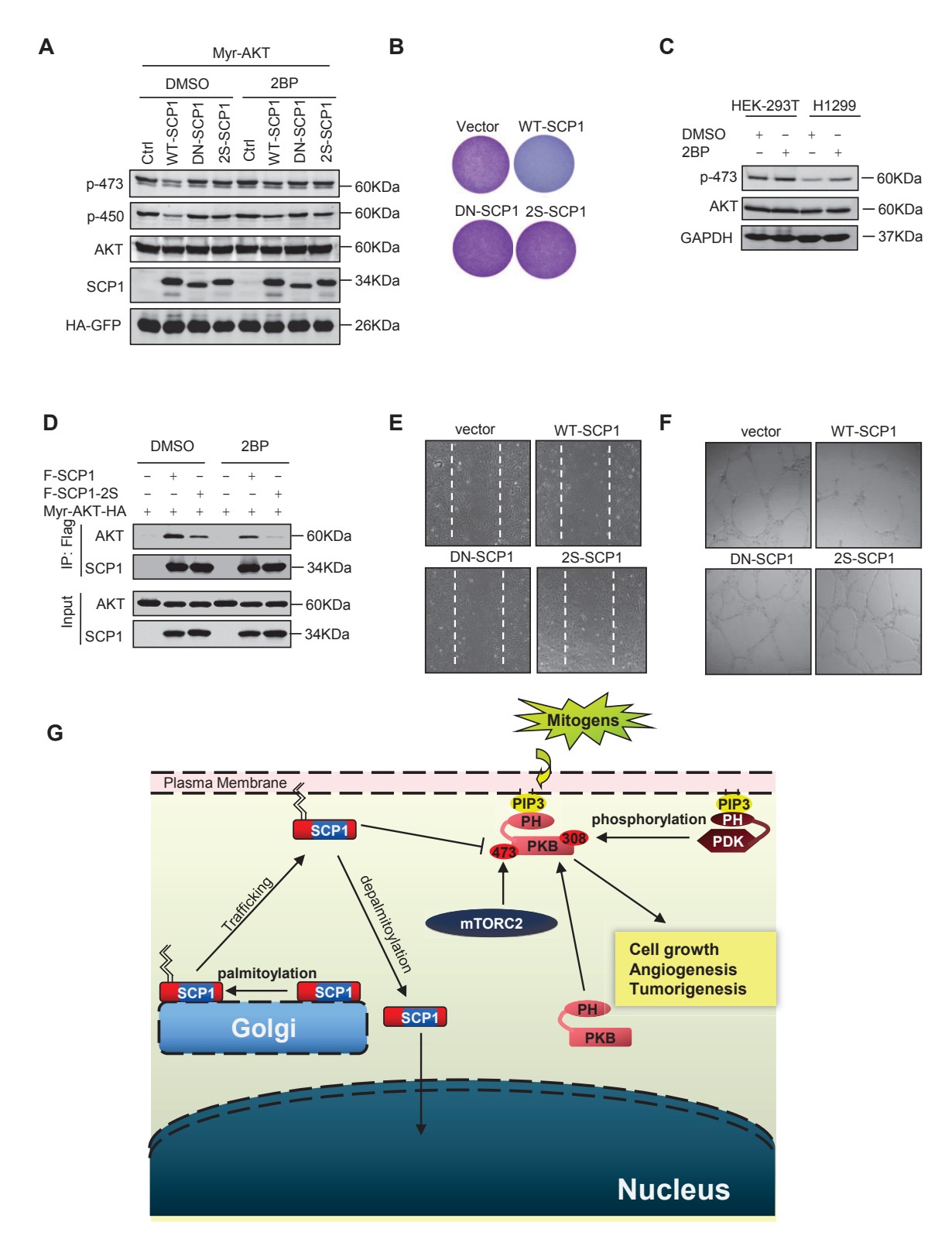

**Figure 6.** Palmitoylation was required for SCP1-mediated AKT inhibition (**A**) Palmitoylation was required for SCP1 to dephosphrylate AKT Ser473. HEK293T cells were transfected with WT-SCP1, DN-SCP1, and 2S-SCP1 for 24 h and treated with DMSO or 2-bromopalmitate (2BP; 10 μM) for 6 h. The phosphorylations of p-Ser473-AKT and p-Thr450-AKT were detected using western blotting. (**B**) Palmitoylation of SCP1 at C44 and C45 was required for its suppression of cell proliferation. Values represent mean ± SD (n = 3) (**C**) Palmitoylation inhibition increased the phosphorylation levels of

*Figure 6 continued*

endogenous AKT Ser473. HEK293T cells and H1299 cells were treated with DMSO or 2BP (10 μM) for 6 h. The phosphorylation of p-Ser473-AKT was detected using western blotting. (D) Depalmitoylation blocked the interaction between SCP1 and AKT. HEK293T cells were transfected with Myr-AKT-HA and FLAG-SCP1 or FLAG-SCP1-2S for 24 h and treated with DMSO or 2BP (10 μM) for 6 h. The interaction between AKT and SCP1 or SCP1-2S was detected using western blotting. (E) SCP1 blocked HUVEC migration in a palmitoylation-dependent manner. HUVECs were transfected with vector, WT-SCP1, DN-SCP1, and 2S-SCP1, respectively. Cell migration was detected using a wound healing assay. (F) SCP1 blocked HUVEC tube formation in a palmitoylation-dependent manner. HUVECs were transfected with vector, WT-SCP1, DN-SCP1, and 2S-SCP1, respectively. Tube formation was detected using a tube formation assay. (G) The working model for SCP1 palmitoylation and cell membrane localization is shown.

The following figure supplement is available for figure 6:

**Figure supplement 1.** SCP1 suppresses AKT-mediated biological functions.

depalmitoylation can regulate protein function and subcellular localization (*Kang et al., 2008*; *Fukata and Fukata, 2010*). Due to its special nature, palmitoylation may be particularly important for modulating protein function during cycles of activation and deactivation through its effect on a protein's affinity for membranes, subcellular localization, and interactions with other proteins. Moreover, reversible palmitoylation of signaling proteins allows proteins to rapidly shuttle between intracellular membrane compartments, since this palmitate cycling can be regulated by different physiological stimuli, which contributes to cellular homeostasis. Thus, the identification of a novel palmitoylated protein like SCP1 could be critical for clarifying special points of cross-talk between molecular signals that are implicated in diverse cellular functions. Based upon our findings, it would be reasonable to assume that the distribution of SCP1 at the plasma membrane via its N-terminal palmitoylation might intensively deal with some uncovered functions of SCP1. We first looked for direct evidence of SCP1 palmitoylation. From a variety of pharmacological inhibitor tests, substantial evidence confirmed the existence of SCP1 palmitoylation and further clarified a distinct role of palmitoylation, but not of farnesylation nor prenylatin, in the control of the membrane recruitment of SCP1 (*Figure 2* and *Figure 2—figure supplement 1*). Our results bridge the gap between the SCP1 post-translational modification and biological function observed in a previous study by Martin and Cravatt, who identified SCP1 as a palmitoylation substrate in their search for palmitoylated proteins (*Martin and Cravatt, 2009*). Thus, we conclude that SCP1 is a palmitoylated phosphatase.

From our data, SCP1 palmitoylation is indeed a reversible process that is dynamically regulated by palmitoylation/depalmitoylation (*Figure 2H*). Palmitoylation of SCP1 stabilizes the association of SCP1 with membranes, thereby facilitating its vesicular trafficking to the plasma membrane and the subsequent dephosphorylation of AKT by their direct interaction (*Figure 5H and J*). Meanwhile, depalmitoylation releases depalmitoylated SCP1 into the cytoplasm, allowing its return to the Golgi for another round of palmitoylation or its direct nuclear entry, where it activates different downstream signaling cascades (*Figure 6G*). Thus, such regulation of the subcellular localization of SCP1 by reversible palmitoylation/depalmitoylation may intimately link it to multiple intracellular signaling pathways. Accordingly, our data prove that the blockade of palmitoylation significantly induces the nuclear localization of SCP1, indicating that SCP1 can shuttle between the plasma membrane and nucleus, which is consistent with previous reports that SCP1 can dephosphorylate nuclear oncoproteins such as Snail, c-Myc, and Smad (*Knockaert et al., 2006*; *Wu et al., 2009*; *Wang et al., 2016*), suggesting that such a novel phenotype of SCP1 shuttling observed in our study may be closely correlated with oncogenesis and tumor progression. In addition, palmitoylation has been shown to be regulated by different signaling molecules such as nitric oxide or Src kinases (*Aicart-Ramos et al., 2011*). It will be instructive to investigate whether the SCP1 subcellular localization is regulated under certain physiological or pathological conditions and whether such conditions could also regulate SCP1 palmitoylation-mediated signaling.

In order to explore the role of SCP1 palmitoylation in the regulation of biological processes, we conducted a variety of experiments including both in vitro and in vivo studies. Our data indicate that the palmitoylation of SCP1 is essential for AKT dephosphorylation at the plasma membrane, whereas such palmitoylation is not required for its phosphatase activity (*Figure 6* and *Figure 6—figure supplement 1*). Meanwhile, AKT dephosphorylation by SCP1 requires a direct interaction between them at the plasma membrane (*Figure 6*). This is consistent with several lines of evidence showing a tight

correlation between AKT activity and its interaction with some specific phosphatases. For example, AKT binds to APPL1 in the endosome, where it regulates the specificity for its substrates (*Schenck et al., 2008*; *Saito et al., 2007*). Another AKT phosphatase, PHLPP1, is also membrane associated via binding to scribble, which is necessary for its dephosphorylation of AKT (*Hung and Link, 2011*). Furthermore, we discovered that C44/C45 cysteines are indispensable for SCP1 palmitoylation and subsequent direct interaction with AKT (*Figures 5* and *6*). Such an intimate link between membrane-localized SCP1 by palmitoylation on C44/C45 cysteines and the dephosphorylation of AKT is highlighted by embryonic vasculogenesis tests and hind limb ischemia models in SCP1-KO mice with markedly enhanced angiogenesis (*Figure 3*). In addition, SCP1 deficiency-promoted angiogenesis is also intimately involved in tumorigenesis (*Figure 3*). Using EGF/insulin-induced MEF cells derived from WT and SCP1-KO littermates, our data illustrate a direct interaction and consequent dephosphorylation of AKT by SCP1 (*Figure 5* and *Figure 5—figure supplement 1*). Since it is relatively easy and straight-forward to examine or quantify membrane-localized SCP1 together with its interaction with AKT, which is the prerequisite for screening potential inhibitors of AKT activity and consequently fine-tuning AKT signaling during tumorigenesis, our study may open up a brand-new avenue for the exquisite regulation of AKT activity that is anticipated to play a critical role in angiogenesis and tumorigenesis.

To further clarify the underlying mechanisms of SCP1-mediated AKT inhibition and the resultant suppression of angiogenesis and tumor growth, we point out that AKT Ser473 is the key site for SCP1-mediated dephosphorylation (*Figures 4*, *5* and *6*). AKT signaling is regulated through its dephosphorylation on serine/threonine residues in both the cytosol and nucleus (*Stronach et al., 2011*; *Testa and Bellacosa, 2001*; *Lee et al., 2015*). Among these phosphorylation/dephosphorylation sites, Ser473 has drawn much attention recently, as it is assumed to participate in oncogenesis, drug resistance and the anti-apoptosis competence of various types of tumor cells (*Stronach et al., 2011*; *Wendel et al., 2004*). The unique phenotype of AKT Ser473 dephosphorylation endowed by its direct interaction with SCP1 at the plasma membrane sheds light on novel therapy strategies for the modulation of aberrant AKT signaling and consequent tumorigenesis (*Figure 3* and *Figure 6—figure supplement 1*).

In this context, we propose a de novo signaling mechanism such that SCP1 is recruited to the membrane by its palmitoylation at the N-terminus, where SCP1 negatively regulates AKT kinase activity, followed by impaired angiogenesis and tumorigenesis. Considering the importance of AKT signaling in angiogenesis and oncogenic development, we present substantial evidence for the identification of SCP1 as a new membrane-localized phosphatase for AKT that carries obvious elements of novelty and interest to the cancer community.

# Materials and methods

## Plasmids and cell culture

The expression plasmids SCP1, SCP2, SCP3, and SCP4 were the gift from Professor Xinhua Feng (Zhejiang University). SCP1, SCP2, SCP3, and SCP4 were then sub-cloned into pcDNA3.1 vectors with a FLAG, HA, or GFP tag. The deletion mutants of SCP1 were cloned into pcDNA3.1 vectors with a FLAG or GFP tag. All constructs were confirmed by DNA sequencing.

HEK293T (American Type Culture Collection, RRID: CVCL-0063), HeLa (American Type Culture Collection, RRID: CVCL-0030), MDCK (RRID: CVCL-0422), COS7 (RRID: CVCL-0224), MCF7 (RRID: CVCL-0031), and DLD1 cells (RRID: CVCL-0248) were cultured in Dulbecco's modified Eagle's medium (DMEM) (Gibco). H1299 cells (RRID: CVCL-0060) were cultured in 1640 medium (Gibco). PC-3 cells (American Type Culture Collection, RRID: CVCL-0035) were cultured in F-12K medium (Gibco), supplemented with 10% heat-inactivated fetal bovine serum (FBS) at 37°C in 5% $CO_2$. SaOS-2 (American Type Culture Collection, RRID: CVCL-0548) cells were cultured in McCoy's 5a Medium, supplemented with 15% FBS at 37°C in 5% $CO_2$. The identification of all of the cell lines has been authenticated by the American Type Culture Collection through STR (Short Tandem Repeat) profiling. No mycoplasma contamination was detected in the cultured cells.

Transfections were performed using calcium phosphate-DNA coprecipitation for HEK293T cells and SunbioTrans-EZ for HeLa cells (Shanghai Sunbio Medical Biotechnology Co., Ltd). H1299 or

HUVEC cells (RRID: CVCL-2959) were transfected with siRNA oligonucleotides using Lipofectamine 2000.

## Antibodies, reagents, immunoprecipitation and western blotting

Immunoprecipitation and western blotting were performed as in our previous report (*Liu et al., 2010*). Briefly, cells were transfected and lysed using 2× RIPA (Radio Immunoprecipitation Assay) buffer (Tris-HCl, pH 7.4 [100 mM]; NaCl [300 mM]; 1% NP-40; 2% sodium deoxycholate; 10 mM NaF; and 10 mM Na vanadate). The cell lysates were cleared by centrifugation and incubated with 1 µg antibody for 1 h at 4°C followed by incubation with 15 µl protein A and G beads (Santa Cruz) for 2 h at 4°C. Immunoprecipitates were subjected to western blot. For western blot analysis, cells were scraped from the dishes into the lysis buffer. A total of 25 mg of total protein was separated by SDS-PAGE and blotted with Pan Anti-palmitoylation (from Haojie Lu), anti-$\beta$-actin (Santa Cruz 47778, RRID: AB-626632), anti-AKT (Abcam 1085–1, RRID: AB-562034), anti-p473-AKT (CST 4060S, RRID: AB-2315049), anti-p308-AKT (CST 2965S, RRID: AB-2255933), anti-p450-AKT (CST 9267S, RRID: AB-823676), anti-FLAG (Sigma F1804, RRID: AB-262044), anti-HA (Santa Cruz 805, RRID: AB-631618), anti-GFP (Santa Cruz 9996, RRID: AB-627695), anti-GSK3 (Abcam 2199–1, RRID: AB-991733), anti-pS9-GSK3 (Abcam 2435–1, RRID: AB-1267179), and anti-AKT-substrate (CST 9611S, RRID: AB-330302). 2-BP (Sigma), palmostatin B (Merck), EGF (Sigma), insulin (ThermoFisher), CHX (Sigma), FTI-277 (Selleckchem), GGTI-298 (Selleckchem), BFA (Sigma), and AKT inhibitor angiogenesis MK2206 (Selleckchem) were also used in experiments..

## Short hairpin RNA design

Negative control siRNA: 5'-UUCUCCGAACGUGUCACGUU-3'; h*Ctdsp1* siRNA: 5'-GCCGGUUGGG UCGAGACCUU-3'; hPHLPP siRNA: 5'-GGAATCAACTGGTCACATT-3'; Scramble shRNA: 5'-TTC TCCGAACGTGTCACGTTT-3';*Ctdsp1* shRNA: 5'- AGCGACGTCCTCACGTGGATGAGTTCTAG TGAAGCCA CAGATGTAGAACTCATCCACGTGAGGACGC-3'.

## Phosphatase activity assay (pNPP phosphatase assay)

SCP1 was immunoprecipitated from cell lysate after transfection for 24 h. After three washes with reaction buffer (*Wang et al., 2016*), the purified SCP1 was incubated with 5 µl pNPP (50 mM, NEB) in a 50-µl reaction volume adjusted for the conditions of the reaction buffer for 30 min at 30°C. The reactions were then terminated using 1 ml of 1 M NaOH and the absorbance at 405 nm was measured.

## In vitro dephosphorylation assay

AKT was immunoprecipitated from HEK293T cells. The immunoprecipitates were washed three times with lysis buffer and once with phosphatase reaction buffer (50 mM Tris-HCl, pH 6.8, 150 mM NaCl, 10 mM MgCl$_2$, pH 8.0, 10 mM DTT) without phosphatase inhibitors. The immunoprecipitates were then resuspended in the phosphatase reaction buffer and divided into four equal aliquots, three of which were incubated with GST, GST-WT-SCP1, or GST-DN-SCP1. After 10 min at 37°C, the dephosphorylation reactions were terminated and the samples were analyzed by western blotting (*Kops et al., 2002*; *Peng Liao, 2017*).

## Kinase assay

A recombinant GST-GSK3$\beta$ protein was purified from *E. coli* using standard protocols. Flag-AKT1 or its mutant forms were expressed in HEK293T cells and purified using anti-Flag M2 beads (Sigma) for immunoprecipitation. The kinase assay was performed as described in our previous study (*Liu et al., 2010*).

## ABE method

The ABE method was performed as previously described (*Noritake et al., 2009*). Briefly, cells were washed with phosphate-buffered saline containing 10 mM N-ethyl-maleimide (NEM) twice and solubilized with 0.3 ml of lysis buffer (LB; 50 mM Tris-HCl, pH 7.5, 5 mM EDTA, and 50 mM NaCl) containing 1% SDS and 10 mM NEM before harvest. After 30 min of extraction, LB with 1% Triton X-100 and 10 mM NEM was added to a final volume of 1 ml and incubated for 30 min at 4°C. After

centrifugation at 12,000 $g$ for 10 min, the supernatants were precipitated by the chloroform–methanol (CM) method. Precipitated protein was solubilized in 0.2 ml SB (50 mM Tris-HCl, pH 7.5, 5 mM EDTA, and 4% SDS) containing 10 mM NEM at 37°C for 10 min. The protein was diluted into 0.8 ml LB with 0.2% Triton X-100 and 1 mM NEM and incubated overnight at 4°C. NEM was removed by three sequential CM precipitations. Precipitated protein was solubilized in 0.2 ml of buffer SB, and then 0.8 ml HB (1 M hydroxylamine, pH 7.5, 150 mM NaCl, 0.2% Triton X-100, and 1 mM biotin-HPDP) or buffer TB (1 M Tris-HCl, pH 7.5, 150 mM NaCl, 0.2% Triton X-100, and 1 mM biotin-HPDP) was added. The mixture was incubated for 1 h at room temperature and subjected to CM precipitation. The precipitated protein was dissolved in 0.2 ml SB, diluted into 0.8 ml LB containing 150 mM NaCl, 0.2% Triton X-100, and 200 µM biotin-HPDP, and incubated for 1 h at room temperature.

## In vitro wound healing assay

Confluent HUVECs grown in 12-well plates were treated with MMC (10 mg/ml) for 6 h in order to inactivate cell proliferation. The cells were wounded and images were captured after 12 h.

## Ethical statement

Mice were caged in groups of five in a laminar airflow cabinet under specific pathogen-free conditions, fed with sterilized food and water, and kept on a 12-h light–dark cycle. We checked the bodyweight of the mice every day and observed their drinking and eating conditions, as well as their activity, in order to monitor their health before they were sacrificed. No mice was observed to be ill or dead during the experimental term. No early euthanasia/humane endpoints for animals were performed since none of animals became severely ill/moribund during the experiment(s). Mice were sacrificed by $CO_2$ euthanasia to minimize the suffering of the mice. All treatments were administered according to the Guide for the Care and Use of Laboratory Animals (Eighth Edition). All of the animals were handled according to approved Institutional Animal Care and Use Committee protocols (AR20130902) of the East China Normal University.

## Animal procedures

SCP1-KO mice were generated using CRISPR-CAS9 methods (*Qiu et al., 2013*). Briefly, guide RNA (gRNA) expression vectors were constructed for pGS3-T7-gRNA. The sequence of gRNA is CCCTC TTCTGCTGTGTCTGC. The pGS3-T7-gRNA vector and the Cas9-encoding plasmids were linearized using DraI and NotI, respectively. The linearized templates were transcribed in vitro via run-off reactions using T7 RNA polymerase, the In vitro Transcription T7 Kit (Takara), and the Sp6 mMESSAGE mMACHINE Kit (Ambion). TE solution containing 25 ng/µl gRNA and 50 ng/l Cas9 mRNA was injected into the cytoplasm of one cell-stage embryos. A mismatch-sensitive T7E1 assay was used to identify the founders. To confirm the modifications in the founders, the PCR products from each founder were generated using the TA cloning kit (Takara) according to the manufacturer's instructions. PCR was used to identify the genotype of the offspring from the intercrossed *Ctdsp1*$^{+/-}$ mice.

## Generation of MEF cells

The SCP1-KO MEF cells were generated as described below. Briefly, mice homozygous for *Ctdsp1* were intercrossed. The pregnant female mice were sacrificed at day 13 post-coitum. The individual embryos were collected, and any extra-embryonic tissue was removed. Then, the embryos were dispersed using scissors, and the dispersed tissues were trypsinized at 37°C for 30 min. Trypsin was inactivated by adding DMEM. The cells were isolated via centrifugation at 1000 rpm in a microcentrifuge for 5 min at room temperature. Then, the cells were resuspended in DMEM and were seeded on 10-mm dishes.

## Imaging of whole-mount retinas

Retinas collected from *Ctdsp1*$^{-/-}$ mice and control littermates at age P5 were dissected, fixed, and permeabilized in Tris-buffered saline, 1% bovine serum albumin, and 0.5% Triton X-100 at 4°C overnight. They were then incubated in Alexa 488-conjugated isolectin B4 (*Bandeiraea simplicifolia*; Invitrogen) at 4°C overnight. After five washes, the flat-mounted retinas were analyzed by fluorescence microscopy (*Lee et al., 2014*).

## Mouse model of unilateral hind limb ischemia

The study protocols were approved by the Institutional Animal Care and Use Committee. We used a mouse model of angiogenesis, in which the entire left femoral artery and vein were excised surgically. When hind limb ischemia was induced, new blood vessels grew into the ischemic limb. We prepared this model in $Ctdsp1^{-/-}$ mice and WT mice to determine whether ischemia-induced angiogenesis was affected by the deficiency of SCP1. In brief, mice were subjected to unilateral hind limb ischemia under anesthesia with sodium pentobarbital (50 mg/kg intraperitoneally). Before surgery, bodyweight and systemic arterial blood pressure (SBP) were determined. SBP was determined using a tail-cuff pressure analysis system (TK370C, Unicom) in the conscious state. Capillary angiogenesis and hind limb blood flow were examined by the methods below (*Egami et al., 2006*).

## Laser Doppler blood flow analysis

We measured the ratio of the ischemic (left)/normal (right) hind limb blood flow by laser Doppler blood flowmetry (moorLDI, Moor Instruments). At seven predetermined time-points (before surgery and at postoperative days 1, 3, 7, 14, 21, and 28), we performed laser-beam scanning over the legs and feet. The average laser Doppler blood flowmetry of the ischemic and non-ischemic hind limbs was then computed. To minimize variations as a result of ambient light, blood flow was expressed as the ischemic (left)/normal (right) hind limb laser Doppler blood flowmetry ratio (*Nilsson et al., 1980*).

## Cancer model by xenograft

$2 \times 10^5$ LLCs were injected into $Ctdsp1$-WT or -knockdown littermates. The diameter of the tumor was measured every 2 days, and the volume of the tumor was calculated. At 14–16 days after injection, the tumors were permitted to grow to 15 mm in any direction by the Institutional Animal Care and Use Committee at Massachusetts General Hospital and collected, fixed overnight with 4% paraformaldehyde, embedded in paraffin, and sectioned. Tumor volume was measured by digital caliper every other day, and was calculated using the following formula: tumor volume = length $\times$ width$^2$ $\times$ 0.52. Sections of the tumors were stained with hematoxylin and eosin and analyzed histologically. The angiogenesis in the tumors was analyzed using immunohistochemistry by CD31 staining (*Standiford et al., 2011*).

## Aortic ring sprouting assay

Aortae from 2-month-old WT and $Ctdsp1^{-/-}$-deficient mice were dissected and cut into 1-mm long pieces. Aortic rings were placed in growth factor-reduced Matrigel (BD Biosciences) and cultured for 5 days in EBM-2 medium (Lonza). Images of individual aortic explants were taken and the microvascular sprouting areas were quantified by measuring the area covered by outgrowth of the vascular sprouts with Image J (RRID: SCR-003070) (*Wang et al., 2013*).

## Assay of in vitro capillary tube formation

HUVECs (RRID: CVCL-2959) were cultured in 2% FBS/DMEM cultured on 24-well plates coated with growth factor-reduced Matrigel (BD Biosciences) at $1.5 \times 10^5$ cells per well and were stimulated with VEGF (100 ng/ml). The capillary tube length was measured 16 h after the stimulation (*Wang et al., 2013*).

## Statistical analysis

The significance of differences was determined using the Student t-test. All quantitative data are expressed as means ± SD. $p < 0.05$ was regarded as a significant difference.

## Acknowledgements

We thank Dr. Dianqing Wu for critical reading of the manuscript. We thank Dali Li, Meizhen Liu, and other members of the Wang laboratory for their technique assistance. We thank Dr. Caiyun Fang and Dr. Haojie Lu for providing the Pan Anti-palmitoylation Antibody to us. This work was supported by grants from the National Basic Research Program of China (973 program 2012CB910404), the National Natural Science Foundation of China (91519322, 91440104, 31222037, 31171338,

81502559 81402417, 81502559, 31501141, and 31401217), and the Doctoral Fund of the Ministry of Education of China (20130076110022).

---

## Additional information

### Funding

| Funder | Grant reference number | Author |
|---|---|---|
| National Natural Science Foundation of China | 31501141 | Peng Liao |
| National Natural Science Foundation of China | 31401217 | Rui Wang |
| Doctoral Fund of the Ministry of Education of China | 20130076110022 | Ping Wang |
| National Basic Research Program of China | 2012CB910404 | Ping Wang |
| National Natural Science Foundation of China | 91519322 | Ping Wang |
| National Natural Science Foundation of China | 91440104 | Ping Wang |
| National Natural Science Foundation of China | 31222037 | Ping Wang |
| National Natural Science Foundation of China | 31171338 | Ping Wang |
| National Natural Science Foundation of China | 81402417 | Ping Wang |
| National Natural Science Foundation of China | 81502559 | Xin Ge |

The funders had no role in study design, data collection and interpretation, or the decision to submit the work for publication.

### Author contributions

PL, Data curation, Formal analysis, Investigation, Methodology, Project administration; WW, Data curation, Formal analysis, Investigation, Visualization, Methodology, Project administration; YL, Data curation, Formal analysis, Investigation, Visualization, Methodology; RW, Data curation, Formal analysis, Validation, Investigation, Visualization, Methodology; JJ, Data curation, Software, Formal analysis, Validation, Investigation, Visualization, Methodology, Project administration; WP, Investigation, Visualization, Methodology; YC, XW, Data curation, Investigation, Methodology; MS, Formal analysis, Investigation, Methodology; DJ, JP, Resources, Investigation, Methodology; ML, Resources, Methodology, Project administration; XL, X-HF, Resources, Methodology; PW, Resources, Supervision, Funding acquisition, Writing—original draft, Project administration; XG, Conceptualization, Resources, Supervision, Funding acquisition, Validation, Writing—original draft, Writing—review and editing

### Author ORCIDs

Xin Ge, http://orcid.org/0000-0001-9233-2691

### Ethics

Animal experimentation: Mice were caged in the groups of five in a laminar airflow cabinet under specific pathogen-free conditions, fed with sterilized food and water, and kept on a 12-hour light-dark cycle. We checked the bodyweight of the mice every day, and observed their drinking and eating condition as well as their activity to monitor their health. We checked the bodyweight of the mice every day and observed their drinking and eating condition as well as their activity to monitor their health before they were sacrificed. No mice was observed to be ill or dead during the experimental term. No early euthanasia/humane endpoints for animals was performed since none of animals became severely ill/moribund during the experiment(s). Mice were sacrificed by Carbon

Dioxide Euthanasia to minimize suffering of mice. All treatments were administered according to the "Guide for the Care and Use of Laboratory Animals" (Eighth Edition). All of the animals were handled according to approved institutional animal care and use committee (IACUC) protocols (AR20130902) of the East China Normal University.

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
