## [Decision Letter]

Thank you for submitting your article "Palmitoylated SCP1 is targeted to the plasma membrane and negatively regulates angiogenesis" for consideration by *eLife*. Your article has been favorably evaluated by Tony Hunter (Senior Editor) and three reviewers, one of whom is a member of our Board of Reviewing Editors. The reviewers have opted to remain anonymous.

The reviewers have discussed the reviews with one another and the Reviewing Editor has drafted this decision to help you prepare a revised submission.

Summary:

This manuscript investigates the function of the SCP1 phosphatase and its newly identified palmitoylation PTM. The manuscript makes several key observations that appear to be novel and significant. First, SCP1 is largely membrane localized and this is dependent on palmitoylation of two Cys residues in the N-terminus. Second, SCP1 appears to dephosphorylate the C-terminal phosphate of Akt (Ser473) when SCP1 is membrane localized. Third, the dephosphorylation of Akt inhibits cell growth (vascular, tumor), consistent with an expected effect of inhibition of Akt activity. These three points add significantly to our understanding of SCP1 and Akt function and regulation, and could inform new therapeutic studies. They also run counter to the prevailing view that SCP1 is predominantly an RNA Pol II CTD phosphatase. The work appears rigorously performed. Though the writing is a bit rough and will need copy-editing, the main points are clear. We support publication in *eLife* after attention is given to the points below.

Essential revisions:

1) The model presented by the authors is interesting, but needs to be further elaborated experimentally. The authors suggest some fraction of SCP1 is cytoplasmic/nuclear. Is there a function for non-membrane associated SCP1? Do known and now readily available depalmitoylation inhibitors affect the levels of SCPI S-palmitoylation and function? Is SCPI S-palmitoylation actually dynamic? This could be addressed by pulse-chase experiments. In any event, more discussion on the potential nuclear/cytoplasmic functions of SCP1 is needed.

2) One of the major findings of this study is the membrane localization of SCP1 (Figure 1 and Figure 1—figure supplement 1). To reach this conclusion, the authors did multiple experiments, in a variety of cell lines, by using ectopic SCP1 expression. To complement these overexpression experiments, they did examine endogenous SCP1 in 293T cells (Figure 1). While the endogenous SCP1 appeared to show a strong membrane distribution, the quality of this particular blot is not as good as other results in the study. To validate that they are indeed looking at endogenous SCP1, it is suggested that the authors conduct an SCP1 RNAi experiment; if the signal in Figure 1 is indeed SCP1, then the signal should disappear/greatly reduce upon SCP1 RNAi.

3) The authors note that a previous phosphatase PHLPP has been implicated in cleaving pS473 from Akt. Is it possible that SCP1 could be regulating PHLPP in some fashion and this is in turn is influencing Akt C-tail phosphorylation? This could be investigated by knocking down PHLPP1/2 in the setting of SCP1 overexpression or overexpressing PHLPP when SCP1 is disabled. Experiments of this type would provide information about the relative importance and potential pathway connections of these two phosphatases.

---

## [Author Response]

*Essential revisions:*

*1) The model presented by the authors is interesting, but needs to be further elaborated experimentally. The authors suggest some fraction of SCP1 is cytoplasmic/nuclear. Is there a function for non-membrane associated SCP1?*

In our previous report, SCP1 dephosphates c-Myc protein at Ser62 and non-membrane SCP1-CC still dephosphates c-Myc. Functionally, SCP1 negatively regulated the cancer cell proliferation (Wang et al., 2016). So, we agree that non-membrane associated SCP1 has a functional role in regulation of c-Myc stability and in suppression of c-Myc transcriptional activity at nucleus, and thus affects cancer cell proliferation.

*Do known and now readily available depalmitoylation inhibitors affect the levels of SCPI S-palmitoylation and function?*

Acyl protein thioesterase 1 (APT1) is a primary depalmitoylating enzyme shown to mediate palmitate removal. Palmostatin B, as an inhibitor of APT1 that suppresses S-palmitoylation was reported (Dekker and Hedberg, 2011). It has been established that there are a variety of proteins such as Ras, PSD-95 can be modified by S-palmitoylation (Kang et al., 2008; Linder and Deschenes, 2007). Also, Cravatt et al. suggested that SCP1 could be a palmitoylation substrate too. In our paper, we have shown the direct evidence about the intimate correlation between SCP1 palmitoylation and its newly identified membrane localization, which plays a central role in suppression of AKT-mediated angiogenesis and tumor growth. To address the reviewer’s question, we treated HEK293T cells expressing Flag-SCP1 with palmostatin B or 2-BP (a general inhibitor of palmitate synthesis and palmitoylation), respectively (Figure 2). Based on our data, we confirmed that palmostatin B directly inhibits depalmitoylation of SCP1, leading to a conclusion that depalmitoylation inhibitor palmostatin B affects SCP1 S-palmitoylation, which may influence SCP1 membrane localization and thus affect AKT-mediated angiogenesis and tumor growth.

*Is SCPI S-palmitoylation actually dynamic? This could be addressed by pulse-chase experiments. In any event, more discussion on the potential nuclear/cytoplasmic functions of SCP1 is needed.*

In order to confirm the SCP1 is actually dynamic, we have performed FRAP assay and time-lapse microscopy with or without palmostatin B (depalmitoylation inhibitor) treatment. We found that the inhibition of depalmitoylation promoted the recovery of SCP1 membrane localization (Figure 7). We also showed that 2-BP decreased the level of SCP1 S-palmitoylation (please see Figure 2 in our paper). In addition, we confirmed that the S-palmitoylation is related to the membrane localization of SCP1 (please see Figure 2—figure supplement 1 in our paper). Furthermore, membrane localization of SCP1 is restored after photobleaching (please see Figure 1—figure supplement 1 in our paper). Therefore, we conclude that the membrane distribution of SCP1 is reversible, which is determined by the dynamic state of SCP1 S-palmitoylation. We have made more discussion on the potential nuclear / cyto-plasmic functions in our revised manuscript (Discussion, third paragraph).

Author response image 1.**DOI:**
http://dx.doi.org/10.7554/eLife.22058.017

*2) One of the major findings of this study is the membrane localization of SCP1 (Figure 1 and Figure 1—figure supplement 1). To reach this conclusion, the authors did multiple experiments, in a variety of cell lines, by using ectopic SCP1 expression. To complement these overexpression experiments, they did examine endogenous SCP1 in 293T cells (Figure 1). While the endogenous SCP1 appeared to show a strong membrane distribution, the quality of this particular blot is not as good as other results in the study. To validate that they are indeed looking at endogenous SCP1, it is suggested that the authors conduct an SCP1 RNAi experiment; if the signal in Figure 1 is indeed SCP1, then the signal should disappear/greatly reduce upon SCP1 RNAi.*

To validate the quality of our SCP1 antibody, we have conducted experiment that we knock down SCP1 in HEK293T cells and analyze SCP1 cell fractionation by western-blotting assay (please see Figure 1 in our paper).

*3) The authors note that a previous phosphatase PHLPP has been implicated in cleaving pS473 from Akt. Is it possible that SCP1 could be regulating PHLPP in some fashion and this is in turn is influencing Akt C-tail phosphorylation? This could be investigated by knocking down PHLPP1/2 in the setting of SCP1 overexpression or overexpressing PHLPP when SCP1 is disabled. Experiments of this type would provide information about the relative importance and potential pathway connections of these two phosphatases.*

To detect whether SCP1 dephosphorylates AKT through the regulation of PHLPP phosphatase activity, we performed double knockdown PHLPP and SCP1 in MEF cells. We found that SCP1 or PHLPP depletion can promote AKT phosphorylation at Ser473. Furthermore, double knockdown SCP1 and PHLPP can exaggerate this effect (Figure 5—figure supplement 1). According to the reviewer’s suggestion, we overexpressed PHLPP1 when SCP1 is disabled, we found that PHLPP1 still can decrease the expression of AKT phosphorylation at Ser473 (Figure 5—figure supplement 1). Thus, the phosphatase function of SCP1 that dephosphorylates AKT is independent of PHLPP activity.